palaeontology, evolution

evolutionary rates, disparity, crocodylomorph, ecomorphology, innovation

**Author for correspondence:**
Thomas L. Stubbs
e-mail: tom.stubbs@bristol.ac.uk

# Ecological opportunity and the rise and fall of crocodylomorph evolutionary innovation

Thomas L. Stubbs[1], Stephanie E. Pierce[2,3], Armin Elsler[1], Philip S. L. Anderson[4], Emily J. Rayfield[1] and Michael J. Benton[1]

[1]School of Earth Sciences, University of Bristol, UK
[2]Museum of Comparative Zoology, and [3]Department of Organismic and Evolutionary Biology, Harvard University, Cambridge, MA, USA
[4]Animal Biology, University of Illinois at Urbana-Champaign, Champaign, IL, USA

TLS, 0000-0001-7358-1051; SEP, 0000-0003-0717-1841; AE, 0000-0001-8673-9591;
PSLA, 0000-0001-7133-8322; EJR, 0000-0002-2618-750X; MJB, 0000-0002-4323-1824

Understanding the origin, expansion and loss of biodiversity is fundamental to evolutionary biology. The approximately 26 living species of crocodylomorphs (crocodiles, caimans, alligators and gharials) represent just a snapshot of the group's rich 230-million-year history, whereas the fossil record reveals a hidden past of great diversity and innovation, including ocean and land-dwelling forms, herbivores, omnivores and apex predators. In this macroevolutionary study of skull and jaw shape disparity, we show that crocodylomorph ecomorphological variation peaked in the Cretaceous, before declining in the Cenozoic, and the rise and fall of disparity was associated with great heterogeneity in evolutionary rates. Taxonomically diverse and ecologically divergent Mesozoic crocodylomorphs, like marine thalattosuchians and terrestrial notosuchians, rapidly evolved novel skull and jaw morphologies to fill specialized adaptive zones. Disparity in semi-aquatic predatory crocodylians, the only living crocodylomorph representatives, accumulated steadily, and they evolved more slowly for most of the last 80 million years, but despite their conservatism there is no evidence for long-term evolutionary stagnation. These complex evolutionary dynamics reflect ecological opportunities, that were readily exploited by some Mesozoic crocodylomorphs but more limited in Cenozoic crocodylians.

## 1. Introduction

Biodiversity is distributed unevenly across time and phylogeny [1,2]. Some groups are morphologically, ecologically and numerically diverse, while their closest relatives are not, and other groups once had great diversity that is now diminished. Understanding these dichotomies is a fundamental aim of evolutionary biology [3]. Many studies have shown that episodic bursts of rapid expansion characterize major evolutionary radiations, when ecological opportunities trigger innovation in new adaptive zones [1–5]. So far, much attention has been on vertebrate groups with extraordinary modern biodiversity, such as birds, mammals, squamate reptiles and teleost fishes [6–9], as researchers seek to understand the origins and drivers of their evolutionary variation and success. Relatively few studies have considered these same questions in groups with low modern biodiversity [10,11], even though modern rarity and conservatism may mask a much richer evolutionary history that is only revealed by the fossil record.

In this context, crocodylomorphs are an incredible group for understanding changing biodiversity [12–16]. Today there are approximately 26 crocodylian species (crocodiles, caimans, alligators and gharials), with limited morphological and ecological diversity as semi-aquatic ambush predators [17]. This low diversity, combined with the geological longevity of modern crocodylian families [18]

(approx. 100 Ma to present), has resulted in crocodylians being labelled as 'conservative' and even 'living fossils' [19]. However, the wider crocodylomorph fossil record reveals 230 million years of past diversity and innovation, particularly in the Mesozoic [20–25], including terrestrial apex carnivores, small herbivores and fast-moving omnivores, and ocean-dwelling fish eaters and macropredators. Adaptation to these divergent modes of life led to major morphological transformations, but only a few quantitative studies have explored large-scale morphological diversification in crocodylomorphs, so far focusing on disparity of the skull [26,27], jaw [22,28] and body size [29–31]. The great morphological and ecological diversity of extinct crocodylomorphs, and the contrast to apparent conservatism in modern representatives, hints at complex and unexplored macroevolutionary dynamics in the group.

There are important unanswered questions about the patterns, tempo and drivers of crocodylomorph morphological evolution. A strong link between higher crocodylomorph species diversity and warmer global temperatures in the past has been documented [12,14,15,32], and some have suggested that crocodylomorph diversity and body size disparity are characterized by sporadic diversifications in particular groups and at particular times, driven by ecological opportunities like habitat and dietary shifts [20,29,31,33]. The well-established links between crocodylomorph morphology and ecology (ecomorphology), particularly skull and jaw shape changes, biomechanics and feeding [34–37], mean that crocodylomorph ecomorphology provides a perfect case study to explore the roles of ecological opportunities in driving and constraining adaptive innovation and evolutionary radiations [17,18].

Here, we present a macroevolutionary study of ecomorphological disparity and evolutionary rates in crocodylomorphs encompassing their 230-million-year history. We first explore disparity by investigating two-dimensional geometric variation in the skull and lower jaw, as proxies for ecomorphological diversity. Next, we reveal temporal trends behind the rise and fall of crocodylomorph disparity, noting a series of evolutionary radiations that characterize the group. Finally, evolutionary rates are quantified to test if habitat and dietary shifts sparked rapid adaptive morphological diversification, and to explore whether crocodylians have different evolutionary dynamics to other, once disparate and now extinct, crocodylomorph clades.

## 2. Methods and material

### (a) Sampling

We sample 240 skulls and 205 lower jaws (see electronic supplementary material, tables S1 and S2). All crocodylomorph species preserving adequate skull and jaw material were included and each sample represented a species, with taxa ranging from the Carnian (approx. 237 Ma) to the Present. Missing species either lacked skull or jaw material, or the specimens were too damaged or distorted. A database of dorsal and lateral images was assembled by photographing museum specimens, from colleagues, or from figures and reconstructions in the literature.

### (b) Shape analyses and morphospace

Variation in dorsal skull shape and lateral jaw shape were used as proxies for ecomorphological disparity [21,22]. Disparity was quantified using two-dimensional geometric morphometrics,

with a mixed landmark/semi-landmarks approach (see electronic supplementary material, figure S1). The skull landmarking regime is a hybrid of those used before [21,26,27], and jaw landmarks are modified from Stubbs *et al.* [22]. In 66 skull samples, landmarks were modified from the coordinate data of Wilberg [26]. In all other skull and jaw samples, landmarks were digitized on images using tpsDig [38]. Generalized Procrustes analyses (GPA) were used to align all landmarks and remove the noise effects of size, position and rotation, in the R package *geomorph* [39]. During the GPA, the semi-landmarks were allowed to slide iteratively to minimize bending energy between each specimen and the average shape. The final sets of aligned landmark coordinates were then subjected to principal components analysis (PCA) in *geomorph*, to visualize morphospace and explore major features of the variation. Thin plate splines were generated to observe shape transformations along major axes (electronic supplementary material, figure S2). Morphospace occupation in subclades and time bins was explored by plotting subsets of the total morphospace.

### (c) Phylogeny

A composite crocodylomorph supertree was assembled following other recent macroevolutionary studies [27–30,40]. The supertree topology is largely based on Godoy *et al.* [29] and modified from the formal supertree of Bronzati *et al.* [20]. We manually added more taxa, guided by published taxonomic and phylogenetic evidence, to maximize coverage and match the landmark data (see detailed description in the electronic supplementary material). The full supertree includes 373 crocodylomorphs and three pseudosuchian outgroup taxa. Uncertainties in interrelationships are reflected by polytomies in the supertree topology. Primary analyses are based on a topology where thalattosuchians are within Neosuchia as sister to Tethysuchia [20], and the gavialids are positioned as sister to tomistomines, rather than 'thoracosaurs', based on evidence from genomic studies and tip-dated Bayesian approaches [41]. Other topologies were tested, and results are consistent with the position of thalattosuchians and gavialids modified based on competing hypotheses [41,42] (see detailed description in the electronic supplementary material, figures S19–S30).

Evolutionary trees were time-calibrated to estimate branch durations. Four time-scaling approaches were used to ensure consistent results: cal3 [43], Hedman [44,45], equal [46] and minimum branch length [46] approaches (see detailed description in the electronic supplementary material). Temporal occurrence data are modified from Godoy *et al.* [29], representing the bounds of species occurrence based on first appearance dates (FADs) and last appearance dates (LADs) (electronic supplementary material, table S4). For each dating approach, 100 time-calibrated trees were generated where polytomies in the supertree were randomly resolved and a single occurrence date for each taxon was sampled from a uniform distribution between their FAD and LAD. Primary analyses are based on the cal3-dated trees and other results are consistent and presented in the supplementary materials (electronic supplementary material, figures S7–S17).

### (d) Temporal disparity trends

Disparity is multifaceted and can reflect the density, spread or overall expanse of morphospaces [47,48]. We use a combination of metrics, measured in 24 stage or multi-stage time

bins, spanning the Late Triassic to Holocene (electronic supplementary material, table S3). We first calculated within-bin mean Procrustes distances (MPD) directly from the aligned landmark data, giving an overview of average dissimilarity through time. To complement this, disparity was quantified using morphospace coordinates from all axes based on the minimum spanning tree length (MST) metric in the R package *dispRity* [48], with partial rarefaction where time bins with a sample size greater than the average of all bins were reduced to the average, and those with sample sizes lower than the average used their original sample. For both MPD and MST, bootstrapping with 1000 iterations was used to generate 95% confidence intervals. Neither MPD nor MST fully encapsulates total morphospace size. Therefore, within-bin morphospace volumes were calculated based on the first three morphospace axes. Raw volumes are susceptible to outliers, so to measure morphospace expanse more effectively we use alpha-shape volumes. A range of alpha values were tested to explore the effects of progressively enveloping points and removing more 'empty' morphospace. Partial disparities for major subclades were calculated to explore their relative contributions to total disparity through time [49].

The aforementioned disparity calculations describe conventional approaches within discrete bins using the empirical sample of skull and jaws. Alternative sub-sampling methods use 'time-slicing', which instead measures disparity at fixed points in time using time-calibrated trees and incorporating estimated ancestral morphologies [50]. We apply this method here to 100 time-calibrated crocodylomorph supertrees, to calculate time-sliced mean pairwise distances, MST and alpha-shape volumes through time at 24 approximately 10 million-year intervals, using the R package *dispRity* [48] and custom code. 'Spaghetti plots' were generated to illustrate all 100 time-sliced iterations for each disparity metric. We also calculated time-sliced partial disparity by using one randomly selected cal-3 dated tree for sampling.

## (e) Evolutionary rates

Rates of morphological evolution were analysed in a Bayesian framework based on the multivariate variable rates model in BayesTraits [8,51]. Significant morphospace axes scores were used as traits (skull PC1–2, jaw PC1–4), determined using the broken stick and Auer-Gervini methods in the R package *PCDimension* [52]. Rate heterogeneity was analysed with a Bayesian Markov chain Monte Carlo (MCMC) reversible-jump algorithm for the 400 time-scaled crocodylomorph trees (100 for each dating approach). Each analysis was run for 2 billion iterations, parameters were sampled every 80 000 iterations and the first 400 million iterations were discarded as burn-in. To detect shifts in evolutionary rates, the multivariate variable rates model rescales branches where the variance of trait evolution differs from that expected in a homogeneous (Brownian motion) model [51]. These 'rate scalars' are estimated for each branch and represent the amount of evolutionary acceleration or deceleration relative to the background rate (heterogeneity). Stepping-stone sampling, with 1000 stones and 100 000 iterations per stone, was used to calculate the marginal likelihood of two models (heterogeneous versus homogeneous rates) and model fit was compared using Bayes factor tests. The smallest effective sample size (all ESS > 200) was used to confirm run convergence in the

R package *CODA* [53]. The 'variable rates post processor' tool was used to extract the final parameter values.

Rates results were summarized by generating consensus trees and quantifying rates through time. In all 100 time-scaled iterations of each dating approach, the branches in the time-scaled trees were replaced by the mean rate scalar parameters, and then a consensus tree was computed using the R package *phytools* [54] and plotted with *ggtree* [55]. Rates through time were calculated using the variable rates post processor in 1 Myr time bins per tree and accounting for shared ancestry [56]. We explore mean rates across the analysed trees in all crocodylomorphs, and then in major monophyletic subclades of interest, including Notosuchia, Tethysuchia, Thalattosuchia and Crocodylia.

# 3. Results and discussion

## (a) Crocodylomorph disparity and ecomorphology

Patterns in crocodylomorph skull morphospace are similar to those identified in previous analyses that have used comparable sampling and landmarking [21,26,27], with the majority of shape variation being captured by just two principal axes (figure 1*a*; electronic supplementary material, figures S2 and S3). PC1 (76.1%) reflects transformations in snout length and width, and overall skull robusticity, ranging from taxa with slender skulls and massively elongated narrow rostra to broad skulls with short and wide snouts. PC2 (13.1%) represents variation in snout shape, with taxa varying from broad-snouted 'duck-faced' and 'pug-faced' forms to those where the snout is slender, and the margins are mediolaterally compressed and sometimes 'domed' (oreinirostral). PC2 also reflects the position of the orbits, which may be located either towards the sagittal midline of the skull roof or towards the lateral margins of the skull (figure 1*a*; electronic supplementary material, figure S2).

These changes in skull shape have important biomechanical consequences, with potential functional and ecological relevance. Robust skulls with broad snouts have high resistance to bending and torsion and experience lower stress when biting [34–36]. Having short snouts reduces the effective biting area and is associated with specialized and selective feeding modes, and often with heterodont teeth. This is common in notosuchians [25], basal crocodyliforms [23] and alligatoroids like *Hassiacosuchus* and *Gnatusuchus* [57] (figure 1*c*). Slender skulls and elongated snouts experience more stress, limiting prey options, but have greater hydrodynamic efficiency when biting in water and an increased relative biting area for capturing small and fast-moving prey [34–36,58]. This morphology evolved many times convergently and represents a textbook example of a selective regime driven by a common diet in thalattosuchians, tethysuchians, many neosuchians (e.g. goniopholidids), gavialids and crocodylids (figure 1*c*) [14]. Having relatively long snouts also reduces the proportional area for muscle attachment in the post-orbital portion of the skull, leading to generally weaker bites relative to the size of the skull [28,59], and the opposite is true for taxa with short snouts and larger post-orbital areas, which accommodate more jaw-closing musculature [23,60]. Laterally positioned orbits are common in terrestrial and fully marine crocodylomorphs and provide a different field of vision when compared to the dorsally positioned orbits of many ambush hunting semi-aquatic crocodylomorphs, where orbits positioned higher on the skull

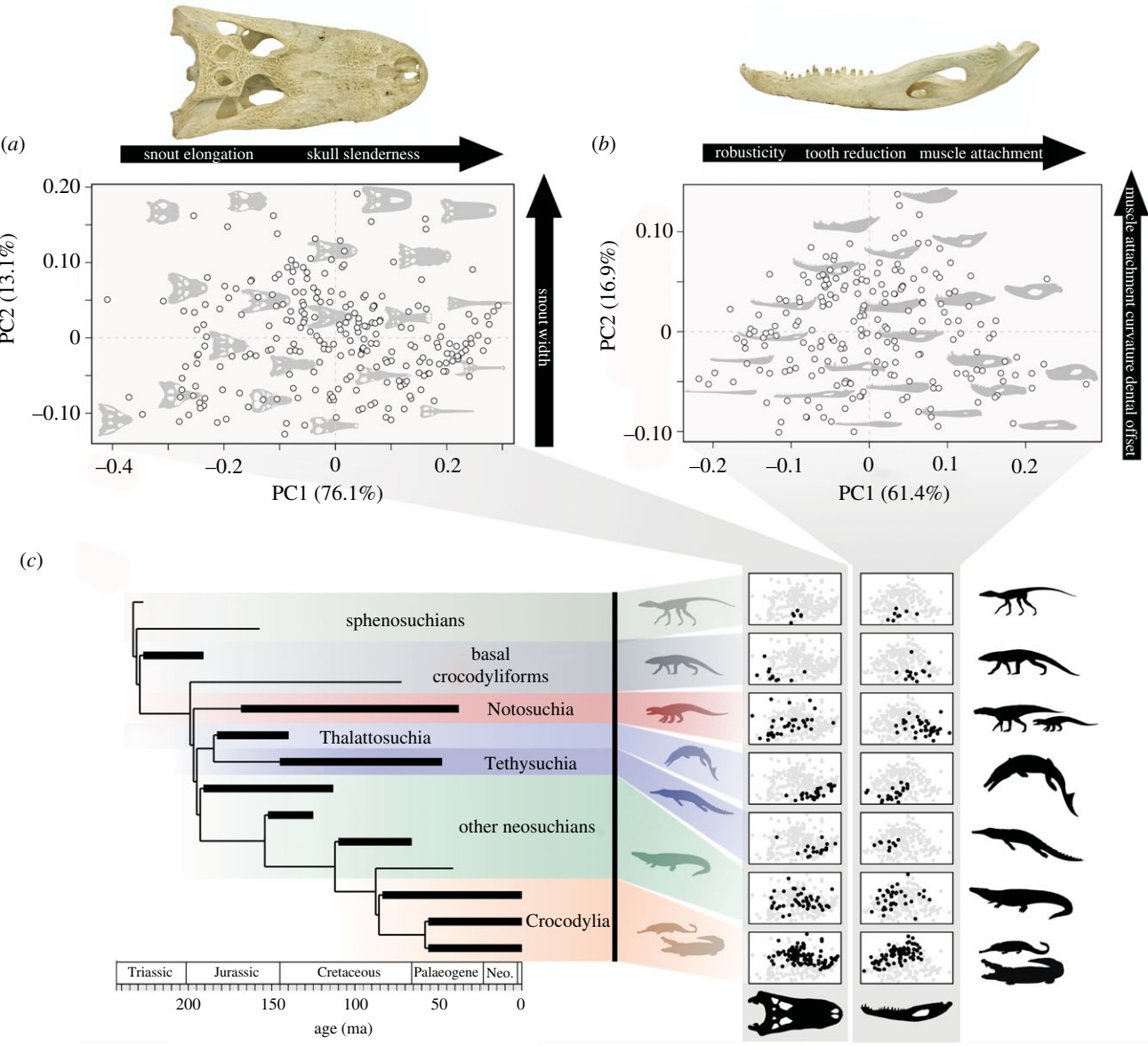

**Figure 1.** Crocodylomorph morphological disparity. (*a*) Skull shape morphospace based on PC1 and PC2. (*b*) Jaw shape morphospace based on PC1 and PC2. Skull and jaw morphologies are plotted showing the distribution of forms and major shape changes are labelled. (*c*) Simplified crocodylomorph phylogeny plotted alongside skull and jaw PC1–PC2 morphospaces showing the distribution of major clades (black circles) in the context of all crocodylomorphs (grey circles). (Online version in colour.)

roof aid vision when the animal is almost totally submerged in the water [61].

Crocodylomorphs have incredible lower jaw disparity [22,23], and shape variation is also encapsulated succinctly in a small number of principal axes (figure 1*b*; electronic supplementary material, figures S2 and S3). PC1 (61.4%) shows variation in the relative size of the tooth-bearing region compared to the post-dentary area of muscle attachment, changes to the robusticity of the dentary, overall jaw depth and the dorsoventral orientation of the retroarticular process. PC2 (16.9%) reflects changes in the position of the jaw joint relative to the tooth occlusal plane, the relative posteromedial flaring of the angular and festooning of the dentary. PC3 (6.2%) represents changes in the dorsoventral orientation of the retroarticular process and the overall curvature of the jaw, while PC4 (4.9%) shows variation in retroarticular process length and compression of the dentary (figure 1*b*; electronic supplementary material, figure S2).

These jaw shape innovations also have clear biomechanical implications. Slender jaws have less hydrodynamic resistance, whereas elongated tooth rows provide a larger surface for puncturing prey when biting—both beneficial when snapping fast-moving prey in water [58,62], and common in thalattosuchians, tethysuchians, neosuchians and crocodylians such as gavialids, *Crocodylus* and *Euthecodon* (figure 1*c*). However, slender jaws and dentaries have reduced resistance to bending, torsion and stresses during feeding [28,62], and increasing the relative length of the tooth row reduces the potential size for post-dentary muscle attachment [28,59,60]. Deeper, more robust jaws can better resist stresses when consuming harder prey, and are often associated with reduced tooth rows in specialist herbivorous, omnivorous and carnivorous terrestrial notosuchians and basal crocodyliforms (figure 1*c*), some of which had complex masticatory systems or heterodont teeth and have often been described as 'mammal-like' crocodylomorphs [23,35,63]. The position of the jaw joint relative to the occlusal plane has major implications for the relative size of

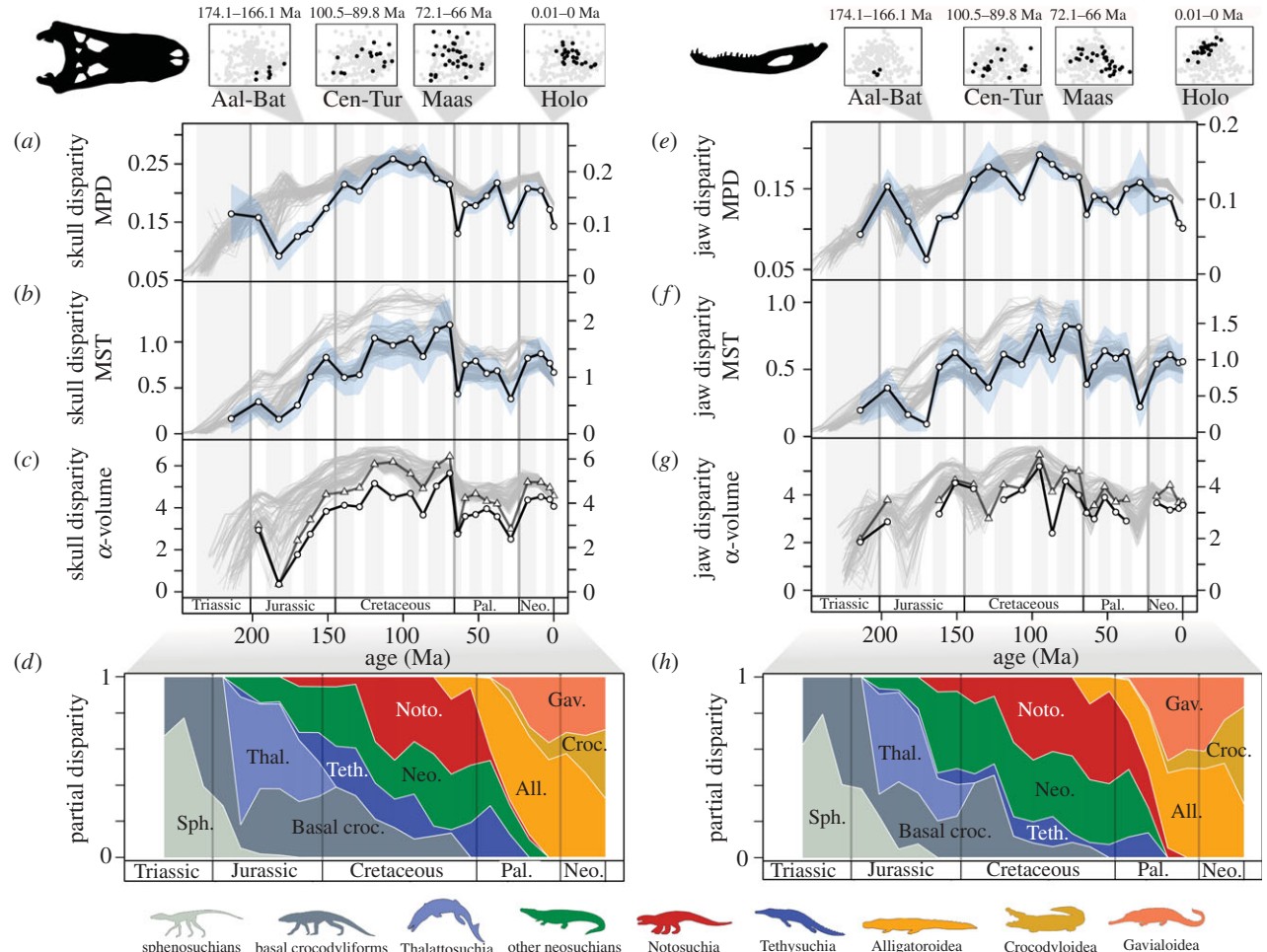

**Figure 2.** Temporal patterns of crocodylomorph morphological disparity. Disparity in the skull (*a*–*c*) and jaw (*e*–*g*) is plotted in 24 time bins (white circle, black line) based on within-bin mean Procrustes/pairwise distances (MPD) (*a,e*), minimum spanning tree lengths (MST) (*b,f*), and morphospace alpha-shape volumes (*c,g*). Blue envelopes in (*a*), (*b*), (*e*) and (*f*) show 95% confidence intervals after bootstrapping with 1000 iterations. In (*c*) and (*g*), the two solid black line curves represent alpha-shape volume disparity for two alpha parameter values, with points in morphospace more tightly enveloped in the lower curves. At the top, the distribution of taxa in skull and jaw morphospace (black circles) is plotted in four time bins (left to right, Aalenian–Bathonian, Cenomanian–Turonian, Maastrichtian and Holocene), illustrating times of low, moderate and high disparity. All time bins are plotted in the supplement (electronic supplementary material, figures S5 and S6). For both the skull and jaw, and for all three disparity metrics, 'spaghetti plots' with grey lines in all panels illustrate the results from 100 iterations of phylogenetic time-sliced disparity. Partial skull (*d*) and jaw (*h*) disparity of crocodylomorphs from the Late Triassic to Holocene based on time-sliced sampling from a single tree, at 24 approximately 10-million-year time slices (empirical partial disparity plotted in electronic supplementary material, figure S35). (Online version in colour.)

the moment arms of major muscle groups. Larger offsets notably increase the moment arms of the major adductor muscles facilitating more powerful bites, and the opposite is true for reduced offsets [23]. Large occlusal offsets are often combined with increased flaring of the angular, providing more area for muscle attachment on the mandible [60] (figure 1*b*). Both these traits are common in generalist crocodylids, alligators, caimans and many extinct neosuchians like allodaposuchids (figure 1*c*), but are even more pronounced in taxa with crushing bites and often bulbous posterior teeth [23,57]. Among living crocodylians, this morphology is seen in the Chinese alligator (*Alligator sinensis*) and broad-snouted caiman (*Caiman latirostris*), but it is best exemplified by the Late Cretaceous neosuchian *Iharkutosuchus* and extinct alligatoroids like *Allognathosuchus*, *Brachychampsa* and *Gnatusuchus* (figure 1; electronic supplementary material, figure S4). Retroarticular process length and orientation are also pivotal to bite force, as it is the site of insertion for the two most massive muscles and acts as a significant anatomical in-lever in crocodylomorphs [37]. Elongated retroarticular processes are also linked to enhanced jaw-opening speed in aquatic dyrosaurids, thalattosuchians and gavialids [17].

## (b) Cretaceous disparity maximum and Cenozoic decline

Temporal patterns of crocodylomorph skull and jaw disparity (figure 2) show the Cretaceous was a prolonged interval of great morphological variation [22,26,27]. During the Cretaceous morphospace was expansive, taxa were widely distributed, and there was high dissimilarity. Pairwise dissimilarity was generally greatest between the Aptian and Santonian, and then decreased in the Campanian and Maastrichtian; however, the spread of taxa and overall morphospace volume was greatest in the Campanian and Maastrichtian when morphospace was most widely explored (figure 2; electronic supplementary material, figures S5 and S6). Notosuchians radiated in the Cretaceous and expanded their contribution to overall crocodylomorph morphospace, making up 40–60% of disparity during the Cretaceous high disparity plateau (figure 2*d,h*). There are many bizarre and morphologically outlying crocodylomorphs known from the middle to Late Cretaceous, most notably notosuchians like *Malawisuchus*, *Anatosuchus*, *Libycosuchus*, *Comahuesuchus*, *Simosuchus* and *Mahajangasuchus*, but also basal crocodyliforms such as *Zosuchus*, and neosuchians like *Iharkutosuchus*, *Stomatosuchus* and *Deinosuchus* (electronic

supplementary material, figures S3 and S4), which also greatly contribute to high disparity (figure 2d,h). Alpha-shape volumes, which mitigate against the effect of outliers, confirm that the Cretaceous was a time of expansive morphospace occupation, followed by a marked reduction in the Cenozoic (figure 2c,g).

Crocodylomorphs had lower disparity prior to the Cretaceous, particularly during the Late Triassic to Middle Jurassic, when crocodylomorph morphospace comprised mainly sphenosuchians, basal crocodyliforms and, in particular, thalattosuchians (figure 2). The application of phylogenetic time-slicing [50] (figure 2, grey lines) and incorporation of estimated ancestral morphospace increases disparity in this interval relative to the empirical within-bin data. Sampling in the late Early to Middle Jurassic is dominated by marine thalattosuchians, which occupy a limited area at the extremes of morphospace, with slender jaws and longirostrine skulls (figure 1). The fossil record of Jurassic sphenosuchians, basal crocodyliforms, early neosuchians and potential notosuchians is poor [64,65], and there are fewer samples to include in disparity analyses. These poorly represented groups in the late Early to Middle Jurassic can be modelled by including estimated ancestral nodes in morphospace; the phylogenetic time-slicing approach expands their disparity contributions (figure 2d,h) and raises overall disparity to levels closer to those of the Late Jurassic and Cenozoic (figure 2, grey lines). This highlights the importance of incorporating phylogenetic information in disparity analyses to account for poorly sampled lineages [50] and suggests that future discoveries of new Jurassic terrestrial crocodylomorphs could increase overall disparity in this interval.

Crocodylomorph disparity declined across the Cretaceous–Palaeogene (K–Pg) boundary in all metrics and remained consistently low through the Cenozoic and into the Recent (figure 2). The Danian bin, immediately following the K–Pg mass extinction, has a poor fossil record for crocodylomorphs [64], but both 'within-bin' and 'time-sliced' disparity show a decline across the K–Pg boundary, although the scale of decline is marginally lessened by the time-slicing approach (figure 2, grey lines). Previous work has proposed that the K–Pg mass extinction had little impact on crocodylomorph evolutionary history, as extinctions were staggered throughout the Cretaceous (e.g. thalattosuchians, goniopholidids), many higher clades passed through the event (e.g. notosuchians, tethysuchians, crocodylians), and overall crocodylomorph diversity bounced back in the early Palaeogene, driven by the diversification of alligatoroids [13,20]. In the early Palaeogene, crown crocodylians (alligatoroids, crocodyloids and gavialoids) replaced 'non-crocodylian' crocodylomorphs as the dominant, and eventually sole, contributors to disparity (figure 2d,h). Most omnivorous and herbivorous notosuchians became extinct by the end-Cretaceous and notosuchians were only represented by carnivorous sebecosuchians thereafter [25,66]. This undoubtedly contributed to the reduction of total crocodylomorph disparity in the Cenozoic, with fewer crocodylians evolving robust brevirostrine skulls and none with specialist 'mammal-like' jaws.

## (c) Fast evolutionary rates and ecological radiations

Rates of crocodylomorph morphological evolution were highly heterogeneous, with all BayesTraits iterations strongly supporting a variable rates model (minimum log Bayes factors greater than 10 across all time-scaled trees [51]). Fast evolutionary rates were widely distributed across phylogeny (figure 3), and results suggest that novel ecological opportunities, resulting from major habitat and dietary shifts in crocodylomorphs [20,33], catalysed rapid ecomorphological innovation in skull and jaw shape. Notably fast rates are found consistently throughout the disparate terrestrial notosuchians, and also at the base of the wider clade comprising the typically longirostrine marine and semi-aquatic thalattosuchians and tethysuchians, and then within both clades. These three ecologically divergent clades show much faster evolutionary rates through time than the background pooled rates across all crocodylomorphs (figure 3c,d).

High notosuchian disparity in the Cretaceous was associated with very rapid ecomorphological evolution (figure 3). Notosuchians filled multiple ecological niches, had exceptional diversity of feeding modes and contributed massively to overall crocodylomorph disparity [23] (figures 1 and 2). Some enigmatic 'mammal-like' notosuchians, with herbivorous, omnivorous and insectivorous diets [25], show rapid innovation in skull and jaw shape, most notably the rapid evolution of brevirostrine skulls and robust jaw morphotypes in taxa like, Simosuchus, Comahuesuchus, Anatosuchus, Pakasuchus and Malawisuchus (figure 4a; electronic supplementary material, figures S7 and S8). However, fast rates were not just seen in some small 'mammal-like' notosuchians, but instead widely throughout the group, also in larger bodied terrestrial carnivores (e.g. baurusuchids) and even large freshwater semi-aquatic notosuchians with distinct skull and jaw shapes (e.g. mahajangasuchids and Stolokrosuchus) (figures 3 and 4a). Therefore, it was not just the evolution of enigmatic 'mammal-like' forms that contributed to notosuchian success, but continued innovation and expansion within morphospace, and widely distributed fast evolutionary rates throughout most of the Cretaceous. This result is consistent even when using trees that notably increase Cretaceous notosuchian branch lengths and pull early node divergences into the Jurassic (see electronic supplementary material, figures S11–S21).

Thalattosuchians radiated during the Early Jurassic and became major components of Mesozoic marine ecosystems [67]. They rapidly evolved longirostrine skulls and slender, snapping jaws, aiding lateral head movements and hydrodynamic efficiency associated with fish eating [28] (figures 1e and 3). This represents a major trait shift (i.e. [51]), diverging from the skull and jaw morphologies of early diverging terrestrial crocodylomorphs, like sphenosuchians and protosuchians. Moderately fast rates are also seen in metriorhynchid thalattosuchians (figures 3 and 4b), particularly at the base of geosaurines, which later expanded thalattosuchian morphospace and had more robust oreinirostral skulls, deeper jaws and serrated dentitions [68], associated with eating larger bodied prey. Evidently, ecological opportunities in the oceans were enough to drive rapid evolutionary change at the origin, and to a lesser extent during the evolution, of thalattosuchians (figure 4b). Importantly, our results are consistent with alternative phylogenetic placements for Thalattosuchia, a contentious issue in crocodylomorph systematics [29,42]. Fast evolutionary rates are still seen in thalattosuchians when placing them outside Neosuchia as early diverging crocodylomorphs and sister group of all Crocodyliformes (see electronic supplementary material, figures S25–S28, S31–34).

Marine dyrosaurid tethysuchians show a burst of fast morphological evolution in the latest Cretaceous and earliest

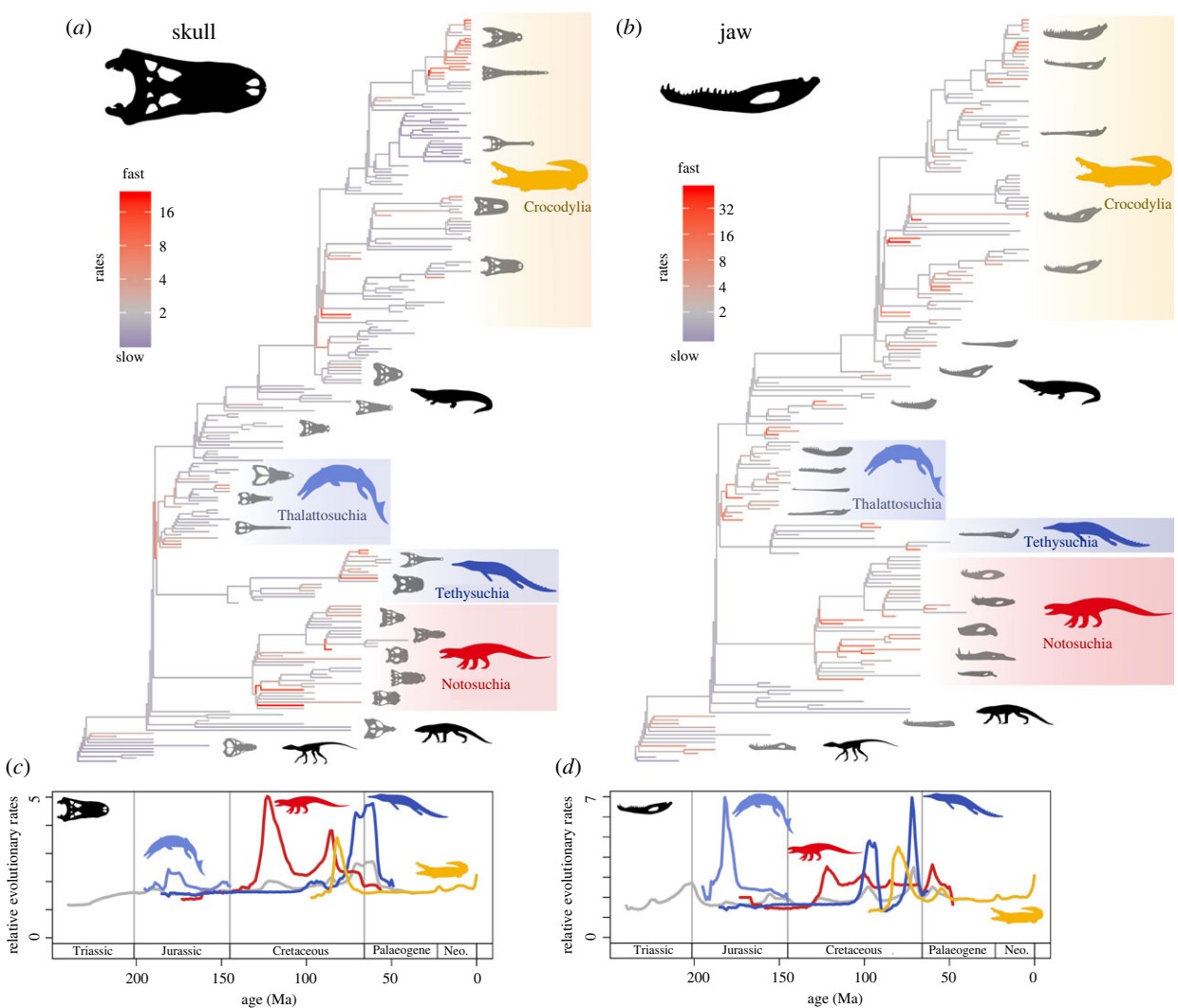

**Figure 3.** Rates of crocodylomorph morphological evolution. Rates of (*a*) skull and (*b*) jaw shape evolution are plotted on time-calibrated phylogenies, based on the outputs from multivariate variable rates models in BayesTraits. Phylogenetic branches are coloured according to the mean rate scalar parameters, derived from the mean of 100 analytical iterations, grading between slow (blue), intermediate (grey) and fast (red) rates. Groups of focus are highlighted: thalattosuchians (blue), notosuchians (red), tethysuchians (dark blue) and crocodylians (yellow). Phylogenies plotted with all taxa labelled are in the electronic supplementary material, figures S7 and S8. Rates of crocodylomorph (*c*) skull and (*d*) jaw shape evolution through time based on 1 Myr time slices are plotted showing major subclades, based on the mean of 100 analytical iterations, showing all crocodylomorphs (grey), thalattosuchians (blue), notosuchians (red), tethysuchians (dark blue) and crocodylians (yellow). Results for individual iterations are illustrated as spaghetti plots in the supplement (electronic supplementary material, figures S9 and S10).

Cenozoic (80–60 Ma), following 100 million years of slow to moderate rates in other tethysuchians (figures 3*c,d*). Dyrosaurids achieved great diversity during this turbulent interval [69], when other competing marine reptile groups became either less diverse in the latest Cretaceous (e.g. poly-cotylid plesiosaurs) or extinct at the K–Pg mass extinction event (all plesiosaurs and mosasaurs). Dyrosaurids mostly had longirostrine skulls with slender snapping jaws, adapted for fish eating [70], but also include rapidly evolved mesorostrine forms and even extremely short-snouted and robustly skulled taxa like the durophagous *Anthracosuchus* [71] (figure 3). A combination of emptied ecospace and ecological diversity may have driven the rapid, albeit short-lived, radiation of dyrosaurids.

Exploitation of ecological opportunities, such as exploring diverse and specialized diets in terrestrial notosuchians, or conquering the marine realm in thalattosuchians and dyrosaurids, have long been considered a major driving force behind evolutionary diversifications [1,4,5]. In crocodylomorphs, these ecological transitions have been linked to marked convergent 'regime shifts' in body size [29] and skull shape [27]. By exploring the lost history of crocodylomorph ecomorphological disparity and evolutionary rates, we show that their macroevolution, in some respects, mirrors trends seen across vertebrates, where large swathes of biodiversity arose from a small number of rapid expansions [2,6,8]. This 'evolution-through-jumps' has been considered a driving force behind the uneven distribution of diversity across time and phylogenetic trees and can be traced back to G. G. Simpson's [1] model of 'quantum evolution'. In crocodylomorphs, much of this disparity was lost through the vicissitudes of Earth's history, and crocodylians are now the sole representatives of this ancient clade.

## (d) Crocodylian rates and evolutionary constraints
For much of the last 80 million years, crocodylians had slow to moderate evolutionary rates, following an initial burst in the Late Cretaceous (figure 3). This initial burst is driven by the

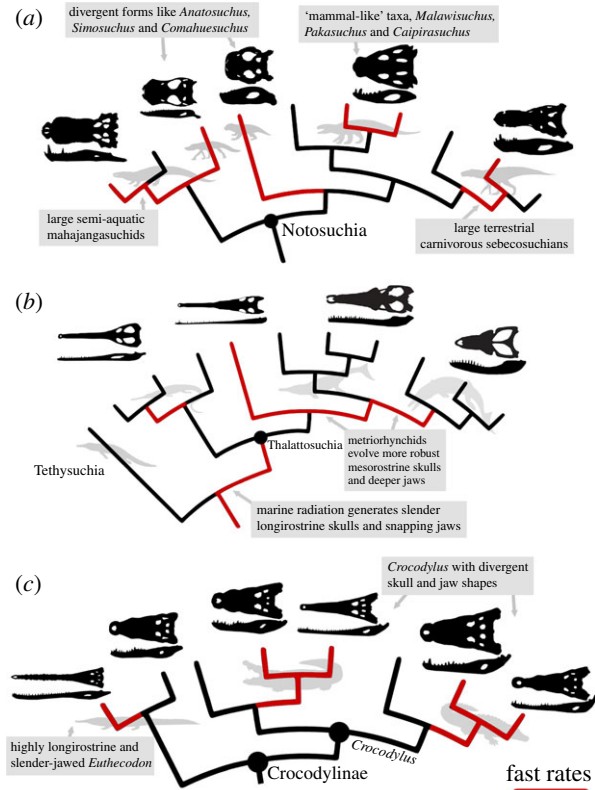

**Figure 4.** Summary illustrating fast rates of ecomorphological evolution in crocodylomorphs. (*a*) notosuchians, (*b*) thalattosuchians and (*c*) crocodylids, consistently show fast evolutionary rates in both skull and jaw shape, across all analytical iterations. Fast rate branches are highlighted in red on simplified phylogenies for the groups showing summarized results from both the skull and jaw. (Online version in colour.)

rapid evolution of distinctive broad-snouted early alligatoroids, like *Deinosuchus* and *Brachychampsa*. Unlike other once diverse Mesozoic crocodylomorphs, crocodylians show no other prolonged fast rate excursions or long intervals with rapid rates. However, there is no evidence for a substantial slowdown in evolutionary rates representing stasis or stabilizing selection (rate scalars < 1 [51]) (figure 3*b*,*d*). Although they are important components of modern ecosystems [72], crocodylians have remained as generalist semi-aquatic ambush predators with medium to large body sizes and have been described as evolutionarily constrained [17]. They do, however, occupy an unexpectedly wide area of total skull and jaw morphospace (figure 1*c*), but this disparity accrued steadily over the last 80 million years with few major evolutionary bursts (figure 3).

This raises questions of why Cenozoic crocodylians did not explore the same breadth of ecomorphospace seen in Mesozoic crocodylomorphs or undergo major expansions within more specialized adaptive zones. Competition from placental mammals may have played a role. Most terrestrial niches, including those exploited by notosuchians, became saturated by mammals throughout the Palaeogene—driven by rapid speciation and high rates of morphological evolution [73]. Similarly, crocodylomorphs were never fully able to exploit marine environments following the extinction of dyrosaurid tethysuchians, perhaps limited by competition from cetaceans that radiated in the Eocene [74]. Climate, combined with intrinsic physiological restrictions, would have also been important. As cold-blooded organisms, the

geographical distributions and diversity of extant and extinct crocodylians have been constrained by environmental temperature [12,14,16]. Climates cooled significantly during the late Palaeogene, limiting crocodylians to warm, tropical, semi-aquatic habitats and restricting their morphological disparity.

Fast evolutionary rates are consistently seen in the extinct highly longirostrine crocodylid *Euthecodon* and living *Crocodylus* (figures 3 and 4*c*). This may, in part, be inflated by intense sampling of the latter genus (13 living species), leading to a high concentration of very short phylogenetic branches in recent crocodylids (figure 3*a*,*b*). Alternatively, it could be a real ecomorphological signal. *Euthecodon* had an exceptionally long snout and very slender jaws compared to the closely related mesorostrine *Brochuchus* (figure 4*c*). Living *Crocodylus* includes closely related taxa that have disparate skull and jaw shapes (figure 4*c*), for example, the slender-skulled Orinoco crocodile (*C. intermedius*) versus the broader-snouted Morelet's crocodile (*C. moreletii*) and Cuban crocodile (*C. rhombifer*), or the slender-snouted freshwater crocodile (*C. johnstoni*) compared to the generalist mesorostrine saltwater crocodile (*C. porosus*). This apparent morphological plasticity may reflect flexible developmental control of craniofacial evolution [75] or may be linked to rapid ecological transitions to fish eating ecologies in the slender-snouted crocodylids [17].

# 4. Conclusion

Our work highlights the importance of ecological opportunity in driving innovation [4], even in a once diverse clade with now diminished biodiversity. Ecologically divergent crocodylomorphs, like thalattosuchians and notosuchians, rapidly filled novel adaptive zones, and then continued to innovate, seemingly not constrained by climate or physiology [76], and evolved divergent skull and jaw morphologies reflecting functional specializations. By contrast, crocodylian disparity accumulated steadily and ecological opportunities were limited for most of the last 80 million years. This provides a textbook example of contrasting evolutionary dynamics within a diverse ancient clade, which could only be revealed through large-scale comparative analyses. A so-called 'living fossil' clade such as the crocodylomorphs may show slow rates in some subclades at certain times, but it had, and has, the potential for fast rates and rapid morphological diversification. It is important to study the macroevolutionary processes responsible for generating biodiversity in ancient clades with low modern diversity [10,11], as well as the highly diverse modern groups frequently investigated.

Data accessibility. All data and code are available from the Dryad Digital Repository: https://dx.doi.org/10.5061/dryad.7sqv9s4rr [77]. The BayesTraits postprocessing tool is available at http://www.evolution.reading.ac.uk/VarRatesWebPP/.

Authors' contributions. T.L.S., S.E.P., E.J.R. and M.J.B. formulated the study. T.L.S. collected data, analysed the data and wrote code, with additions from A.E. and P.S.L.A. T.L.S. drafted the manuscript and figures with input from all authors.

Competing interests. We declare we have no competing interests.

Funding. This work was funded by BETR grant no. NE/P013724/1 and ERC grant no. 788203 (INNOVATION) to T.L.S. and M.J.B., and NERC grant no. NE/L002434/1 to A.E. and M.J.B.

Acknowledgements. We are indebted to many people for providing images of specimens: James Clark, Roger Smith, Gert Wörheide, Oliver Rauhut, Neil Clark, Chris Brochu, Melanie Vovchuk, Juan

Porfiri, Jeremías Taborda, Judith Babot, Hugo Carrizo, Diego Pol, Caitlin Syme, Rodolfo Salas-Gismondi, Lucy Souza, Jeremy Martin, Jessica Cundiff, Thomas Smith, Suresh Singh and the late Jon Tennant and Jaime Powell. We thank Eoin Gardiner for discussions about Cenozoic crocodylomorph evolution. Cranial silhouettes in the figures are by Haley O'Brien, Paul Gignac (used with permission) and T.L.S.; jaw silhouettes are by T.L.S. Body silhouettes are modified from Stubbs *et al.* [22] and Ballell *et al.* [28], or artwork by Deverson da Silva, Nobu Tamura and Dmitry Bogdanov (available at http:// phylopic.org/). This work was carried out using the computational facilities of the Advanced Computing Research Centre, University of Bristol - http://www.bris.ac.uk/acrc/.

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
