## [Peer Review File · Proceedings of the Royal Society B: Biological Sciences]

Review History

RSPB-2021-0069.R0 (Original submission)

Review form: Reviewer 1 (Pedro Godoy)

Recommendation

Accept with minor revision (please list in comments)

Scientific importance: Is the manuscript an original and important contribution to its field?
Marginal

General interest: Is the paper of sufficient general interest?
Good

Quality of the paper: Is the overall quality of the paper suitable?
Excellent

Is the length of the paper justified?
Yes

Should the paper be seen by a specialist statistical reviewer?
No

Do you have any concerns about statistical analyses in this paper? If so, please specify them explicitly in your report.

No

It is a condition of publication that authors make their supporting data, code and materials available - either as supplementary material or hosted in an external repository. Please rate, if applicable, the supporting data on the following criteria.

Is it accessible?

Yes

Is it clear?

Yes

Is it adequate?

Yes

Do you have any ethical concerns with this paper?

No

Comments to the Author

In this manuscript, Stubbs et al. quantify cranial and lower jaw shape variation in Crocodylomorpha using 2D geometric morphometrics. They estimate disparity through time and also make comparisons between different crocodylomorph subgroups. Finally, they calculate rates of morphological evolution, indicating subgroups and moments in which higher rates are observed. The manuscript is very well-written and the figures were nicely drawn. I also consider that the data analysis was also appropriately conducted. Therefore, I do not have many technical comments to make in that sense.

However, I do have a more general comment. Although I agree with most of the authors' interpretations/conclusions, I have to say I am a bit disappointed with the paper. I was expecting to learn something new about crocodylomorph evolution and I am afraid this paper does not provide it. It does involve a few novel analyses (e.g., rates analyses), but it mostly focuses on types of data (skull and lower jaw shape) and analyses (geometric morphometrics and morphological disparity) already explored before for the group. I understand that, for a group like crocodylomorph, it's important to quantify and document large-scale patterns, given that we still have much to learn about the group when compared to other tetrapods (e.g., dinosaurs and mammals). So, I do think this manuscript has its value in building this knowledge. I am just not sure it significantly improves or changes our current understanding of the evolution of the group or provides novel results that furthers the field.

A related issue is that, in the Introduction and mainly in the Abstract, the authors do not make it clear what is being quantified (i.e., cranial and lower jaw shape variation). In the Abstract, in particular, the authors go straight from the "background" to the "results", without mentioning what analyses or methods are used. This might induce the reader to understand that other aspects of "ecomorphology" are being analysed (other than cranial/lower jaw shape) or different methods are being applied (other than 2D geometric morphometrics).

Other very minor comments:

- Is there any reason to limit the disparity calculation to only the first 3 PCs? I believe including more (maybe all) would not be much computationally demanding (at least for analyses done in R).
- It would be interesting to estimate disparity using the sum of variances as the disparity metric. This would allow the comparison with the results of Stubbs et al. (2013), Wilberg (2017), and Godoy (2020).

I have also attached the annotated pdf with very few additional comments. Finally, I am available if authors want to contact me for further discussion.

Sincerely,
Pedro Godoy

Review form: Reviewer 2 (Philip Mannion)

Recommendation

Accept with minor revision (please list in comments)

Scientific importance: Is the manuscript an original and important contribution to its field?

Good

General interest: Is the paper of sufficient general interest?

Good

Quality of the paper: Is the overall quality of the paper suitable?

Good

Is the length of the paper justified?

Yes

Should the paper be seen by a specialist statistical reviewer?

No

Do you have any concerns about statistical analyses in this paper? If so, please specify them explicitly in your report.

No

It is a condition of publication that authors make their supporting data, code and materials available - either as supplementary material or hosted in an external repository. Please rate, if applicable, the supporting data on the following criteria.

Is it accessible?

Yes

Is it clear?

Yes

Is it adequate?

Yes

Do you have any ethical concerns with this paper?

No

Comments to the Author

Dear authors,

This is a welcome and interesting evaluation of the ecomorphological evolution of crocodylomorphs through time. It uses state-of-the-art approaches for calculating and analysing diversification rates and disparity, coupled with an impressive dataset. I've provided comments on an annotated version of the combined PDF of the MS and below summarize a few issues for consideration (some of this is partly duplicated in the annotated comments):

(1) Crocodylomorph diversity trends are described as understudied in the Introduction, but I think this is demonstrably untrue. In addition to several papers already cited in the MS, I've listed four additional papers that aren't currently cited (see annotated comments for details) and these are just restricted to studies evaluating numbers of taxa through time. I think you're fine to make the case that we should study these patterns in crocodylomorphs, regardless that there are other papers. Note that other than this I consider the references in the MS as very up-to-date, with just a few additional papers suggested for inclusion (unusually for a reviewer, none of these are by me!).

(2) There are a couple of instances whereby previous work is framed as an either/or debate, but I don't think that this is really the case. The main instance occurs in the Introduction in which previous work is framed as either presenting a climatic or ecological driver for explaining crocodylomorph diversity patterns, but I don't think this reflects the literature. For example, Reference 12 focuses on temperature, but also discussed the impact of changes in available habitat (e.g. effect of Andean uplift on the proto-Amazonian wetlands) and ecological release (e.g. post-K/Pg opportunism in the marine realm) on crocodylomorph diversity patterns. Similarly, I don't think those that have focused on the role of ecology have discounted climate, so I think it's incorrect to set this up as a dichotomy of competing hypotheses. I'd also recommend incorporating the Solórzano et al. (2020) paper here (currently ref 68), given its relevance to biotic and abiotic impacts on crocodylian diversity.

(3) Is the image database of skulls available somewhere? I realise it's not strictly necessary for being able to replicate your analyses, with the raw data extracted from these images fully available, but it would be a wonderful resource.

(4) Although I like how you've tried to account for poor sampling in some time intervals when calculating disparity, I still think it might be a substantial problem that no method can fully account for though. For example, there's just a long single ghost lineage reconstructed for *Notosuchia* in your SI trees and so presumably your method reconstructs a single ancestral morphology for this back into the Jurassic? However, it seems unlikely that notosuchians just did nothing for tens of millions of years and then popped up on three continents in the Aptian in lots of different forms. The probable notosuchian, *Razanandrongobe*, from the Middle Jurassic of Madagascar, suggests that we're missing a lot of the early diversification of the clade too. There's a similar issue in that thalattosuchians basically just appear in the water fully formed, with no obvious terrestrial 'ancestor'.

These are two of the three clades highlighted as examples of fast evolutionary rates. I wonder how much of this is the result of these clades essentially appearing fully formed and already diverse (the third one is another marine clade too)? Might this be a factor (and potentially a substantial one) in these three clades showing the fastest rates? Going back to *Notosuchia*, that clade is (analytically at least) regarded as having genuinely diversified when they first appear in the fossil record (i.e. the early Late Cretaceous, excluding *Razanandrongobe*), rather than having an unsampled earlier diversification. I find it easier to believe that the overall pattern fits this (i.e. the evidence from the Aptian through to the latest Cretaceous indicates high diversification in terms of new ecomorphologies), but I'd have much less confidence that the 'rise' of notosuchians can be evaluated in this way.

I realise that this is just a problem we can't really deal with, and that diversity-through-time analyses also suffer from genuine vs artefactual absence, but I would suggest expanding on this issue more (particularly when talking about early bursts in these clades) and/or expanding on why the method does a better job of dealing with this issue than I'm assuming.

Best wishes,
Phil Mannion

Decision letter (RSPB-2021-0069.R0)

08-Feb-2021

Dear Mr Stubbs:

Your manuscript has now been peer reviewed and the reviews have been assessed by an Associate Editor. The reviewers' comments (not including confidential comments to the Editor) and the comments from the Associate Editor are included at the end of this email for your reference. As you will see, the reviewers and the Editors have raised some concerns with your manuscript and we would like to invite you to revise your manuscript to address them.

The reviewers and Associate Editor raise concerns of whether the MS is truly novel and appropriate for ProcB, beyond a nice cutting edge methods study, generating more than incremental insights. They would need to be convinced for the paper to be accepted. Furthermore the inaccessibility of original skull image data (not just morphometrics output data) has been raised and this must be rectified if the MS is to be accepted, as per our open science + open data policy; all (not some) data that would enable reproducibility and re-use must be provided.

Research ethics:

Use of animals and field studies:

It is a condition of publication that you make available the data and research materials supporting the results in the article. Please see our Data Sharing Policies (<https://royalsociety.org/journals/authors/author-guidelines/#data>). Datasets should be deposited in an appropriate publicly available repository and details of the associated accession number, link or DOI to the datasets must be included in the Data Accessibility section of the article (<https://royalsociety.org/journals/ethics-policies/data-sharing-mining/>). Reference(s) to datasets should also be included in the reference list of the article with DOIs (where available).

Please submit a copy of your revised paper within three weeks. If we do not hear from you within this time your manuscript will be rejected. If you are unable to meet this deadline please let us know as soon as possible, as we may be able to grant a short extension.

Best wishes,

Dr John Hutchinson, Editor

Associate Editor

Board Member: 1

Comments to Author:

Thank you for your submission to PRSB.

Your manuscript has now been seen by two expert reviewers, both of whom consider your study to be a positive addition to the literature on crocodylomorph macroevolution, and one that is analytically sophisticated. Moreover, they both compliment the clarity of the writing and view the work positively overall.

However, both of them also raise concerns regarding the novelty of the approach and conclusions. Both reviewers note that, while this study goes further than others in some ways, this is the latest of several studies investigating similar topics, and in places the manuscript fails to acknowledge this adequately. As such, the authors should be careful to fully reference the pre-existing literature which sets a precedent for this work, and the second referee has provided a number of suggested additions to the references that will help with this. It seems like an obvious stretch to suggest that the subject matter investigated here is ‘understudied’ – it would be appropriate for the authors to go through the manuscript and ensure that the tone matches the fact that some of the questions investigated here are indeed fairly well studied.

One potential opportunity for this study to set itself apart would be by providing images of the actual specimens underlying the analyses, potentially in the supplement. Fundamentally, this morphometric work is a contribution to our understanding of crocodylomorph morphology, so I think it would be a very useful addition to the manuscript if at least some of the imagery underlying the quantitative work were provided.

As referee 1 suggests, the authors should also improve the clarity with which they discuss exactly which aspects of morphology are the subject of focus at various points in the discussion.

Referee 2 also raises an important question regarding the extent to which results indicating rapid diversification in clades with relatively long ghost lineages may be artefactual, and I think this important point warrants further consideration.

I look forward to seeing a revision of this manuscript, and thank you again for your submission to PRSB.

Reviewer(s)' Comments to Author:

Referee: 1

Comments to the Author(s)

In this manuscript, Stubbs et al. quantify cranial and lower jaw shape variation in Crocodylomorpha using 2D geometric morphometrics. They estimate disparity through time and also make comparisons between different crocodylomorph subgroups. Finally, they calculate rates of morphological evolution, indicating subgroups and moments in which higher rates are observed. The manuscript is very well-written and the figures were nicely drawn. I also consider that the data analysis was also appropriately conducted. Therefore, I do not have many technical comments to make in that sense.

However, I do have a more general comment. Although I agree with most of the authors' interpretations/conclusions, I have to say I am a bit disappointed with the paper. I was expecting to learn something new about crocodylomorph evolution and I am afraid this paper does not provide it. It does involve a few novel analyses (e.g., rates analyses), but it mostly focuses on types of data (skull and lower jaw shape) and analyses (geometric morphometrics and morphological disparity) already explored before for the group. I understand that, for a group like crocodylomorph, it's important to quantify and document large-scale patterns, given that we still have much to learn about the group when compared to other tetrapods (e.g., dinosaurs and mammals). So, I do think this manuscript has its value in building this knowledge. I am just not sure it significantly improves or changes our current understanding of the evolution of the group or provides novel results that furthers the field.

A related issue is that, in the Introduction and mainly in the Abstract, the authors do not make it clear what is being quantified (i.e., cranial and lower jaw shape variation). In the Abstract, in particular, the authors go straight from the “background” to the “results”, without mentioning what analyses or methods are used. This might induce the reader to understand that other aspects of “ecomorphology” are being analysed (other than cranial/lower jaw shape) or different methods are being applied (other than 2D geometric morphometrics).

Other very minor comments:

- Is there any reason to limit the disparity calculation to only the first 3 PCs? I believe including more (maybe all) would not be much computationally demanding (at least for analyses done in R).
- It would be interesting to estimate disparity using the sum of variances as the disparity metric. This would allow the comparison with the results of Stubbs et al. (2013), Wilberg (2017), and Godoy (2020).

I have also attached the annotated pdf with very few additional comments. Finally, I am available if authors want to contact me for further discussion.

Sincerely,
Pedro Godoy

Referee: 2
Comments to the Author(s)
Dear authors,

This is a welcome and interesting evaluation of the ecomorphological evolution of crocodylomorphs through time. It uses state-of-the-art approaches for calculating and analysing diversification rates and disparity, coupled with an impressive dataset. I've provided comments on an annotated version of the combined PDF of the MS and below summarize a few issues for consideration (some of this is partly duplicated in the annotated comments):

(1) Crocodylomorph diversity trends are described as understudied in the Introduction, but I think this is demonstrably untrue. In addition to several papers already cited in the MS, I've listed four additional papers that aren't currently cited (see annotated comments for details) and these are just restricted to studies evaluating numbers of taxa through time. I think you're fine to make the case that we should study these patterns in crocodylomorphs, regardless that there are other papers. Note that other than this I consider the references in the MS as very up-to-date, with just a few additional papers suggested for inclusion (unusually for a reviewer, none of these are by me!).

(2) There are a couple of instances whereby previous work is framed as an either/or debate, but I don't think that this is really the case. The main instance occurs in the Introduction in which previous work is framed as either presenting a climatic or ecological driver for explaining crocodylomorph diversity patterns, but I don't think this reflects the literature. For example, Reference 12 focuses on temperature, but also discussed the impact of changes in available habitat (e.g. effect of Andean uplift on the proto-Amazonian wetlands) and ecological release (e.g. post-K/Pg opportunism in the marine realm) on crocodylomorph diversity patterns. Similarly, I don't think those that have focused on the role of ecology have discounted climate, so I think it's incorrect to set this up as a dichotomy of competing hypotheses. I'd also recommend incorporating the Solórzano et al. (2020) paper here (currently ref 68), given its relevance to biotic and abiotic impacts on crocodylian diversity.

(3) Is the image database of skulls available somewhere? I realise it's not strictly necessary for being able to replicate your analyses, with the raw data extracted from these images fully available, but it would be a wonderful resource.

(4) Although I like how you've tried to account for poor sampling in some time intervals when calculating disparity, I still think it might be a substantial problem that no method can fully account for though. For example, there's just a long single ghost lineage reconstructed for Notosuchia in your SI trees and so presumably your method reconstructs a single ancestral morphology for this back into the Jurassic? However, it seems unlikely that notosuchians just did nothing for tens of millions of years and then popped up on three continents in the Aptian in lots of different forms. The probable notosuchian, Razanandrongobe, from the Middle Jurassic of Madagascar, suggests that we're missing a lot of the early diversification of the clade too. There's

a similar issue in that thalattosuchians basically just appear in the water fully formed, with no obvious terrestrial 'ancestor'.

These are two of the three clades highlighted as examples of fast evolutionary rates. I wonder how much of this is the result of these clades essentially appearing fully formed and already diverse (the third one is another marine clade too)? Might this be a factor (and potentially a substantial one) in these three clades showing the fastest rates? Going back to Notosuchia, that clade is (analytically at least) regarded as having genuinely diversified when they first appear in the fossil record (i.e. the early Late Cretaceous, excluding Razanandrongobe), rather than having an unsampled earlier diversification. I find it easier to believe that the overall pattern fits this (i.e. the evidence from the Aptian through to the latest Cretaceous indicates high diversification in terms of new ecomorphologies), but I'd have much less confidence that the 'rise' of notosuchians can be evaluated in this way.

I realise that this is just a problem we can't really deal with, and that diversity-through-time analyses also suffer from genuine vs artefactual absence, but I would suggest expanding on this issue more (particularly when talking about early bursts in these clades) and/or expanding on why the method does a better job of dealing with this issue than I'm assuming.

Best wishes,
Phil Mannion

Author's Response to Decision Letter for (RSPB-2021-0069.R0)

See Appendix A.

Decision letter (RSPB-2021-0069.R1)

01-Mar-2021

Dear Mr Stubbs

I am pleased to inform you that your manuscript entitled "Ecological opportunity and the rise and fall of crocodylomorph evolutionary innovation" has been accepted for publication in Proceedings B. Congratulations!!

Open Access

Corresponding authors from member institutions (<http://royalsocietypublishing.org/site/librarians/allmembers.xhtml>) receive a 25% discount to these charges. For more information please visit <http://royalsocietypublishing.org/open-access>.

Paper charges

Sincerely,

Dr John Hutchinson

Appendix A

Associate Editor

Board Member: 1

Comments to Author:

Thank you for your submission to PRSB.

Your manuscript has now been seen by two expert reviewers, both of whom consider your study to be a positive addition to the literature on crocodylomorph macroevolution, and one that is analytically sophisticated. Moreover, they both compliment the clarity of the writing and view the work positively overall.

TLS - Many thanks for considering our work and we are very happy about the positive response.

However, both of them also raise concerns regarding the novelty of the approach and conclusions. Both reviewers note that, while this study goes further than others in some ways, this is the latest of several studies investigating similar topics, and in places the manuscript fails to acknowledge this adequately. As such, the authors should be careful to fully reference the pre-existing literature which sets a precedent for this work, and the second referee has provided a number of suggested additions to the references that will help with this. It seems like an obvious stretch to suggest that the subject matter investigated here is ‘understudied’—it would be appropriate for the authors to go through the manuscript and ensure that the tone matches the fact that some of the questions investigated here are indeed fairly well studied.

TLS – we thank the reviewers for highlighting this and we have added additional references in the introduction and checked the tone throughout. We do believe our paper makes many novel contributions to our understanding of crocodile-line reptile evolution and wider ideas about the rise and fall and large clades with complex histories.

One potential opportunity for this study to set itself apart would be by providing images of the actual specimens underlying the analyses, potentially in the supplement. Fundamentally, this morphometric work is a contribution to our understanding of crocodylomorph morphology, so I think it would be a very useful addition to the manuscript if at least some of the imagery underlying the quantitative work were provided.

TLS – we strongly support Proc B’s philosophy on open science and would be happy to share the images with colleagues. In this case there are several reasons why providing all images on a public platform is problematic:

1) For the skull sample, 66 taxa (out of 240) are based on landmark coordinate data from Wilberg 2017, so we do not have the original images for these skull samples. This was outlined in the supplementary materials in the first submission, but we now add this information into the methods section of the main text (a similar approach was used by Godoy 2020).

We have images for all other skull and jaw samples, but there are two issues:

2) some taxa photographs were shared by colleagues with access to the material (see list in the acknowledgements). When sharing these images, we assured our colleagues that they would not be published and/or shared further without their permission - seeking permission from all these researchers now may take considerable time.

3) for some taxa we used images from the literature, where high quality figures in the appropriate orientation were published. Sharing these images via Dryad may raise some ownership/copyright issues.

We have tried to make our work as open as possible in all other ways – we provide the raw data, phylogenies and code, all of which are great resources for future researchers (now uploaded to Dryad <https://doi.org/10.5061/dryad.7sqv9s4rr>). Particularly, our code for processing BayesTraits outputs is a massive contribution; most other groups using this approach have not shared the code they use to consolidate outputs and make figures etc.

As referee 1 suggests, the authors should also improve the clarity with which they discuss exactly which aspects of morphology are the subject of focus at various points in the discussion.

TLS – this has been checked.

Referee 2 also raises an important question regarding the extent to which results indicating rapid diversification in clades with relatively long ghost lineages may be artefactual, and I think this important point warrants further consideration.

TLS – we have provided a detailed response to this below. In short, our methodology is robust to this issue for two reasons: (1) we use different time-calibration methods for the phylogeny, two of which don't give this long single ghost lineage, and all methods give the same general rate result and nothing to change our conclusions. These additional tests were presented in the supplement, but we now provide better links to this in the main text, (2) both methods can inherently account for the issue raised by the reviewer. In disparity analyses it is important to capture the major morphotype (s) (in this case in the estimated ancestor) rather than many examples of such estimated ancestors. Our method captures estimated ancestors in the poorly sampled part of their histories (i.e. Jurassic notosuchians), even from a single long ghost lineage. The variable rates model is designed to account for rate heterogeneity in unsampled taxa and incorporates rate variation on internal nodes reconstructed by the trees (Venditti et al. 2011, *Nature*. p. 396). We also run some additional tests just to ensure that the point raised by the reviewer is not a problem and we include the results below.

I look forward to seeing a revision of this manuscript, and thank you again for your submission to PRSB.

TLS – many thanks, we have included all revised materials.

Reviewer(s)' Comments to Author:

Referee: 1

Comments to the Author(s)

In this manuscript, Stubbs et al. quantify cranial and lower jaw shape variation in Crocodylomorpha using 2D geometric morphometrics. They estimate disparity through time and also make comparisons between different crocodylomorph subgroups. Finally, they calculate rates of morphological evolution, indicating subgroups and moments in which higher rates are observed. The manuscript is very well-written and the figures were nicely drawn. I also consider that the data analysis was also appropriately conducted. Therefore, I do not have many technical comments to make in that sense.

TLS – thank you for your insights and positive comments.

However, I do have a more general comment. Although I agree with most of the authors' interpretations/conclusions, I have to say I am a bit disappointed with the paper. I was expecting to learn something new about crocodylomorph evolution and I am afraid this paper does not provide it. It does involve a few novel analyses (e.g., rates analyses), but it mostly focuses on types of data (skull and lower jaw shape) and analyses (geometric morphometrics and morphological disparity) already explored before for the group. I understand that, for a group like crocodylomorph, it's important to quantify and document large-scale patterns, given that we still have much to learn about the group when compared to other tetrapods (e.g., dinosaurs and mammals). So, I do think this manuscript has its value in building this knowledge. I am just not sure it significantly improves or changes our current understanding of the evolution of the group or provides novel results that furthers the field.

TLS – we thank the reviewer for their generally positive assessment. We believe our work makes several novel contributions and we have checked to see that these are clearly outlined in the main text.

1) fast morphological evolution in ecologically divergent crocodylomorphs is shown for the first time – specifically, sustained fast rates over long intervals linked to continued innovation. This may have been predictable based on biological principles (or with the reviewer's great expertise in crocodylomorph evolution), but it has never been tested or shown before.

2) crocodylians do have sporadic fast rates in some individual taxa/small groups, but not such sustained fast evolution – this is the first time this has been shown in crocodylomorph ecomorphology. A paper about body size rates was published in January 2021, while this submission was in review), but it only partially showed this, and no comparisons were made between higher crocodylomorph clades to test for differences, as we do in this paper.

3) crocodylian rates are not exceptionally slow – this goes against the, perhaps outdated, idea that they are living fossils. Even though this idea is generally not supported by modern literature, this is the first paper to directly test it and show that rates of evolution did not stagnate in crocodylians.

4) these trends point to a shift in the tempo/mode of crocodylomorph evolution, from sporadic ecological expansions to a more conservative evolutionary history within the bounds of a generalist predatory ecology. This has never been shown numerically.

These main conclusions/contributions are in addition to more nuanced insights into morphospace, times of greatest disparity and contribution of different clades through time, which add to the current literature.

TLS - The reviewer highlights that other papers have also looked at large-scale disparity in crocodylomorphs (more on the skull, less in the jaw) and we cite the previous work. We do not see this previous work as a negative thing. If anything, it has established that studying large-scale trends of morphological variation in crocodylomorphs has great value and has set out important principles about the links between ecology and morphology and serves as our inspiration. Here we do not spend considerable time discussing the morphospace, clades distributions and things like convergence because these have been the focus of other papers. We briefly discuss them so we can move on to other important questions about ecomorphological disparity, ecology and rates

A related issue is that, in the Introduction and mainly in the Abstract, the authors do not make it

clear what is being quantified (i.e., cranial and lower jaw shape variation). In the Abstract, in particular, the authors go straight from the “background” to the “results”, without mentioning what analyses or methods are used. This might induce the reader to understand that other aspects of “ecomorphology” are being analysed (other than cranial/lower jaw shape) or different methods are being applied (other than 2D geometric morphometrics).

TLS - Thank you for highlight this, it has now been rectified with additional information in the abstract and introduction:

“In this macroevolutionary study of **skull and jaw shape variation....**”

“We first explore disparity in the group by investigating **2-D** geometric variation in the skull and lower jaw, as a proxy for ecomorphological diversity”

Other very minor comments:

- Is there any reason to limit the disparity calculation to only the first 3 PCs? I believe including more (maybe all) would not be much computationally demanding (at least for analyses done in R).

TLS - One of our metrics is calculated directly from the landmark data (MPD). The other two use PC scores. The minimum spanning tree (MST) length metric calculations did use all PC axes, but this was not made clear in the original submission, this has been clarified:

“To complement this, disparity was quantified using morphospace coordinates **from all axes** based on the minimum spanning tree length (MST) metric in the R package dispRity [51].”

The third metric we use is alpha-shape morphospace volumes, for which we did use just three axes. We limited these calculations to just three axes primarily because of the ‘curse of dimensionality’ – this is an issue that can heavily effect volume-based metrics of high dimensionality data. In short, volume calculations from multiple axes tend towards zero and the volume outputs become incredibly small – this causes issues for measuring size of hulls within morphospace. Thankfully, both the skull and jaw shape variation in our analyses are captured succinctly by a small number of axes.

- It would be interesting to estimate disparity using the sum of variances as the disparity metric. This would allow the comparison with the results of Stubbs et al. (2013), Wilberg (2017), and Godoy (2020).

TLS - We initially considered eight disparity metrics calculated from PC scores (the code provided with the paper allows all these to be explored), but eventually we decided to present the MST and volume results, to provide a different insight into disparity. We found that the sum of variance was very similar to mean pairwise distances (MPD) that are calculated directly from the landmarks. We wish to retain the MPD metric because these were the calculations that involved minimal transformations of the original data, have smaller 95% confidence intervals, and are less affected by outliers during bootstrapping compared to the sum of variances.

I have also attached the annotated pdf with very few additional comments. Finally, I am available if authors want to contact me for further discussion.

TLS - Many thanks, we have worked through the minor corrections from the PDF, and these are shown as tracked changes in the attached revision.

Sincerely, Pedro Godoy

Referee: 2

Comments to the Author(s)

Dear authors,

This is a welcome and interesting evaluation of the ecomorphological evolution of crocodylomorphs through time. It uses state-of-the-art approaches for calculating and analysing diversification rates and disparity, coupled with an impressive dataset. I've provided comments on an annotated version of the combined PDF of the MS and below summarize a few issues for consideration (some of this is partly duplicated in the annotated comments):

TLS - thank you for the feedback and positive assessment of our work. We have worked through the attached pdf and made the small tweaks suggested by the reviewer (please see tracked changes manuscript).

We have also addressed some of the bigger points raised by the reviewer, and these are outlined in detail below. Apologies for the lengthy response of some replies, we wanted to provide as much information as possible.

(1) Crocodylomorph diversity trends are described as understudied in the Introduction, but I think this is demonstrably untrue. In addition to several papers already cited in the MS, I've listed four additional papers that aren't currently cited (see annotated comments for details) and these are just restricted to studies evaluating numbers of taxa through time. I think you're fine to make the case that we should study these patterns in crocodylomorphs, regardless that there are other papers. Note that other than this I consider the references in the MS as very up-to-date, with just a few additional papers suggested for inclusion (unusually for a reviewer, none of these are by me!).

TLS - Thank you for highlighting this. We agree that there are still many important things to understand about crocodylomorph evolution, but it was an error to describe crocodylomorph macroevolution as "understudied". We meant compared to other groups like mammals, birds, fish etc. But there have been other important crocodylomorph macroevolution (particularly diversity and skull shape) studies and we do not want to undermine this. We have changed the opening sentence to "**crocodylomorphs are an incredible group for understanding changing biodiversity**", which I hope the reviewer will agree is more appropriate.

TLS - We have also added six new references to the paper, five concerning previous work on crocodylomorph biodiversity (mainly taxonomic diversity). In earlier editing we had cropped some references out to reduce the word count (we already had 70 references), but we agree it is important to cite as widely as possible within the journal's limits.

(2) There are a couple of instances whereby previous work is framed as an either/or debate, but I don't think that this is really the case. The main instance occurs in the Introduction in which previous work is framed as either presenting a climatic or ecological driver for explaining crocodylomorph diversity patterns, but I don't think this reflects the literature. For example, Reference 12 focuses on temperature, but also discussed the impact of changes in available habitat (e.g. effect of Andean uplift on the proto-Amazonian wetlands) and ecological release (e.g. post-K/Pg opportunism in the marine realm) on crocodylomorph diversity patterns. Similarly, I don't think those that have focused on the role of ecology have discounted climate, so I think it's incorrect to set this up as a dichotomy of competing hypotheses.

TLS – we agree and have modified wording where ideas were set out as competing hypotheses. The ideas are now presented as previously studied drivers that have been proposed, rather than hinting at one being a dominant idea or their being mutually exclusive.

I'd also recommend incorporating the Solórzano et al. (2020) paper here (currently ref 68), given its relevance to biotic and abiotic impacts on crocodylian diversity.

TLS - this reference has now been added here in the introduction, rather than later in the results/discussion.

(3) Is the image database of skulls available somewhere? I realise it's not strictly necessary for being able to replicate your analyses, with the raw data extracted from these images fully available, but it would be a wonderful resource.

TLS (note this is copied from the above response to the editor) – we strongly support Proc B's philosophy on open science and would be happy to share the images with colleagues. In this case there are several reasons why providing all images on a public platform is problematic:

1) For the skull sample, 66 taxa (out of 240) are based on landmark coordinate data from Wilberg 2017, so we do not have the original images for these skull samples. This was outlined in the supplementary table in the first submission, but we now add this information into the methods section of the main text (note a similar approach was used by Godoy 2020).

We have images for all other skull and jaw samples, but there are two issues:

2) for some taxa photographs were shared by colleagues with access to the material (see list in the acknowledgements). When sharing these images, we assured our colleagues that they would not be published and shared further without their permission (seeking permission from all these researchers now may take considerable time).

3) for some taxa we used images from the literature, where high quality figures in the appropriate orientation were published. Sharing these images via Dryad may raise some ownership/copyright issues.

We have tried to make our work as open as possible in all other ways – we provide the raw data, new phylogenies and code, all of which are great resources for future researchers. Particularly our code for processing BayesTraits outputs; most other groups using this approach have not shared the code they use to consolidate outputs and make figures.

(4) Although I like how you've tried to account for poor sampling in some time intervals when calculating disparity, I still think it might be a substantial problem that no method can fully account for though. For example, there's just a long single ghost lineage reconstructed for Notosuchia in your SI trees and so presumably your method reconstructs a single ancestral morphology for this back into the Jurassic? However, it seems unlikely that notosuchians just did nothing for tens of millions of years and then popped up on three continents in the Aptian in lots of different forms. The probable notosuchian, Razanandrongobe, from the Middle Jurassic of Madagascar, suggests that we're missing a lot of the early diversification of the clade too. There's a similar issue in that thalattosuchians basically just appear in the water fully formed, with no obvious terrestrial 'ancestor'.

TLS – We thank the reviewer for appreciating our efforts in accounting for poorer sampling during some intervals of crocodylomorph evolution. We believe our methods (both disparity and rates) can mitigate against this and that our results are robust. We hope to outline this below and present some additional calculations to support this. These issues did initially concern us also, which is what inspired us to run time-sliced disparity and four different dating approaches (this meant running the entire rates analyses for the skull and jaw four times, taking considerable time!).

The reviewer highlights the potential issue of a single/few long ghost lineage(s), meaning that only a few ancestral morphologies may be incorporated prior to the first sampled taxa. When calculating time-sliced phylogenetic skull/jaw disparity we used trees based on four dating approaches (25 iterations of each, giving 100 trees in total). In two of these methods, large ghost lineages are produced, and sampled ancestors are found close to the earliest sampled taxon (giving long ghost ranges back to the origin node of the group, rather than breaking up the branches). However, in the other two approaches node divergences are substantially older giving a lot more branching (and ancestral morphologies) prior to the first fossil occurrences. This is illustrated in the two plots below that are snapshots from our supplement (they are consensus trees from Hedman and equal dated trees). In both plots notosuchians are the focus, and you can see that there are branching events occurring in the Jurassic, each of which would be associated with a morphology that is incorporated within the time-sliced disparity calculations (giving ‘notosuchian like’ morphologies in the Jurassic). We used such trees in our calculations presented in figure 2 (grey lines on all panels) and you can see that there is generally a consistent pattern across all time-sliced iterations (even those not based on trees with these older divergences). Similar principles apply for thalattosuchians and tethysuchians.

Figure highlighting notosuchian branching in the Jurassic, arrows point to Notosuchia base and branching, snapshots taken from figures S12 and S16.

These are two of the three clades highlighted as examples of fast evolutionary rates. I wonder how much of this is the result of these clades essentially appearing fully formed and already diverse (the third one is another marine clade too)? Might this be a factor (and potentially a substantial one) in these three clades showing the fastest rates? Going back to Notosuchia, that clade is (analytically at

least) regarded as having genuinely diversified when they first appear in the fossil record (i.e. the early Late Cretaceous, excluding *Razanandrongobe*), rather than having an unsampled earlier diversification. I find it easier to believe that the overall pattern fits this (i.e. the evidence from the Aptian through to the latest Cretaceous indicates high diversification in terms of new ecomorphologies), but I'd have much less confidence that the 'rise' of notosuchians can be evaluated in this way.

TLS - This comment is linked to the above (potential missing early history from fossil record) but more linked to rates and the same principles generally apply. We note that the reviewer has no issue with our results showing fast morphological evolution in Cretaceous notosuchians, but they highlight our usage of the phrase 'rise of notosuchians' when discussing fast rates at that time. We agree that this is poor wording (since notosuchians may have already partly 'risen' before the Cretaceous), so we have modified the text to clarify this **“High notosuchian disparity in the Cretaceous was associated with very rapid ecomorphological evolution”**.

Back to methodological aspects, we believe our results are robust for several reasons:

Firstly, the Variables rates model in BayesTraits is designed to account for inhomogeneity that would occur if unsampled taxa had different rates (either faster or slower than the sampled taxa). The branching events prior to first sampled taxa incorporate morphological changes that lead to these more morphologically distinctive groups. This is a quote from a paper featuring these methods published by the developers of the software (key points in bold):

Venditti et al. (2011, Nature p. 396): *“We infer historical evolutionary patterns in the rate of body size evolution using a phylogeny constructed from extant species 8,9 . Incorporating fossils 3,21–24 in our analysis could reveal details of the patterns of evolution leading to extinct species, and these could even differ from those for extant species. For example, the rate of evolution in extinct groups such as the triconodonts and multituberculates, which fall between the extant monotremes and the rest of the extant mammals, could have a considerably higher (or lower) rate of change than extant groups. **However, we would not expect the historical trends we describe here for extant species to change qualitatively with the addition of extinct groups, even if those extinct groups did show different patterns of change;** and we would not expect the inclusion of fossils to produce generalized early-burst patterns of change. **The reason is that the methodology we use is specifically designed to account for the sort of inhomogeneity that would occur if an extinct group had a distinct rate of change. Thus, if such a fossil group were included, our method would detect the difference in rate and scale in that section of the tree and leave the rates along the remainder of the phylogeny unchanged.**”*

In other words, if we found more Jurassic notosuchians it will add knowledge about that interval, but it will not substantially change the results we already have for the Cretaceous, likewise for the thalattosuchians and any potential missing Triassic forms.

Secondly, by using four tree-dating approaches and running the analyses of each set, we check our main conclusions are robust to potential evolution happening prior to the first occurrences of major groups. As previously outlined, the different time calibration methods give young/older divergence times, thus more or less branching deeper in geological time, and this will impact the rates as the reviewer suggests. All our conclusions are supported when using these four different tree-dating approaches and we go through each one in our supplement, noting any minor differences. We add an additional sentence explaining this and include more links to the main text and supplement at relevant parts. We chose to present the cal-3 results in the main text because this method is considered the most sophisticated. We have added this sentence to more clearly set out our results (and also no discount any potential earlier fast evolution): *“Therefore, it was not just the evolution of enigmatic ‘mammal-like’ forms that contributed to the notosuchian success, but continued innovation and expansion within morphospace, and widely distributed fast evolutionary rates*

throughout most of the Cretaceous. This result is consistent even when using trees that notably increase Cretaceous notosuchian branch lengths and pull early node divergences into the Jurassic (see electronic supplementary materials, figures S11-21).

I realise that this is just a problem we can't really deal with, and that diversity-through-time analyses also suffer from genuine vs artefactual absence, but I would suggest expanding on this issue more (particularly when talking about early bursts in these clades) and/or expanding on why the method does a better job of dealing with this issue than I'm assuming.

Best wishes,
Phil Mannion

TLS - Yes, potential incompleteness of the fossil record does present difficulties in macroevolutionary studies, although more so in studies of taxonomic richness. In studies of morphological disparity, it is often important to sample major morphotypes (this was achieved by our time-slicing disparity) rather than all examples of every morphotype. In addition, our various rates analyses incorporate some evolutionary history/rate changes prior to the first sampled taxa. One thing we would like to clarify is that we do not specifically say that notosuchians, thalattosuchians or dyrosaurids undergo 'early bursts' and we have also checked through the paper to make sure it is also clear, and there is not mention of early bursts. The EB model would strictly predict a spike in fast rates/high disparity early in the history of a group followed by large drop in rates/disparity. This is not what we see in these groups and it is not important to the conclusions of our study. We instead show that during times of high disparity contributions by these groups they were evolving fast (bursts of rapid evolution, but not necessarily at the start of their history), and in the case of notosuchians and thalattosuchians they had sustained fast rates over long periods and continued to innovate. So, because of this, potentially missing early records of the clades is less important to our results and major conclusions.

TLS - Because we wanted to ensure that the points raised by the review do not impact our major conclusions, we decided to run some ancillary tests. The reviewer's suggestions about potentially missing fossil records and ghost lineages made us want to test "what if we included some hypothetical new taxa". Notosuchians, with *Razanandrongobe* existing significantly before other notosuchians, present the best test case to see what if there were more notosuchians samples from earlier in the clade's history (I actually wonder if *Razanandrongobe* is a notosuchian – but this is a different matter). To explore this, we added *Razanandrongobe* to our supertree along with 5 other hypothetical notosuchians, in random positions. We tested placing these 'hypothetical notosuchians' in both basal and derived positions within Notosuchia. These were assigned ages based on stages from the Middle Jurassic to earliest Cretaceous (see below).

	A	B	C
1	taxa	FAD	LAD
2	Razanandrongobe	168.3	166.1
3	Notosuchia_test_1	163.5	157.3
4	Notosuchia_test_2	157.3	152.1
5	Notosuchia_test_3	168.3	166.1
6	Notosuchia_test_4	157.3	152.1
7	Notosuchia_test_5	145	139.8

We then generated new trees including these 6 new OTU's using the cal-3 and equal dating methods, generating 50 trees for each method and with taxa added in basal/derived positions (200 new trees). The two plots below are samples from the 200 generated trees (left is equal tree with

hypothetical notosuchians placed basally and right is a cal-3 tree with hypothetical notosuchians placed in more derived positions). You can see that this gives more notosuchian branching in the Jurassic, in some trees bringing the origin of all notosuchian subclades into the Jurassic, and also massively increasing the branch lengths of many Cretaceous terminal branches (this is perhaps more unlikely than rapid branching and subclade origins in the Cretaceous?).

Figure showing new node placements with hypothetical notosuchians and *Razanandrongobe* added, blue nodes are all notosuchians, light blue are older nodes in the Jurassic. You can see there are now more notosuchians nodes in the Jurassic, even in the cal-3 trees (left).

Next, we simulated some morphological data by sampling PC scores from our empirical skull data. We sampled PC scores from within the skull morphospace convex hull of known notosuchians (this presumes that *Razanandrongobe* and the hypothetical forms don't fall outside the range of known notosuchians). We now had all the ingredients to repeat our analyses – both phylogenetic time-sliced disparity and evolutionary rates.

	A	B	C	D
1 taxa		PC1	PC2	PC3
2 Razanandrongobe		0.00490037	-0.0206453	0.0
3 Notosuchia_test_1		-0.0983006	-0.0130798	0.0
4 Notosuchia_test_2		-0.1088876	0.10470892	0.0
5 Notosuchia_test_3		-0.1809976	-0.0337379	0.0
6 Notosuchia_test_4		-0.2561609	-0.070993	0.0
7 Notosuchia_test_5		-0.2130912	-0.0423081	0.0
8				
9				

Results – time sliced disparity.

The plots below show the skull morphospace data that is incorporated within these exploratory analyses from two iterations (out of 200 in total). The grey points are all crocodylomorphs (i.e. Fig 1a from main text), red triangles are known notosuchian skulls, black circles are the random simulated morphospace points for *Razanandrongobe* and the 5 hypothetical forms. The time-sliced disparity method incorporates estimated ancestral morphologies, and these are shown here – the blue dots are all notosuchian ancestral morphologies and the light blue are taxa that are now placed in the

Jurassic (matching the trees from above, note that they would be positioned differently in each iteration).

When we rerun time-sliced disparity calculations, now based on 200 iterations with the added ‘test notosuchians’, the results are very consistent with our main results (figure 2 in the main text). The blue lines are mean pairwise distances (2a) and grey lines are minimum spanning tree lengths (2b). Disparity is fractionally increased in the Jurassic when including these hypothetical forms, but the differences are very small. All our main conclusions are robustly supported, and our primary analyses capture potential notosuchian morphotypes from the Jurassic.

We agree that future notosuchian discoveries could provide new insights in their early evolution, and we have added a sentence to the main text results to acknowledge this: **“This highlights the importance of incorporating phylogenetic information in disparity analyses to account for poorly sampled lineages [50], and suggests that future discoveries of more Jurassic terrestrial crocodylomorphs could increase overall disparity in this interval.”** However, we think that our discovery of highest disparity in the Cretaceous is very robust.

Results – rates

Next, we tested some rates analyses with this simulated data and trees. Due to time limitations (analyses would take several weeks) we ran analyses based on 50 cal-3 trees with the ‘test’ notosuchians placed basally, aiming to pull-back many notosuchians nodes far into the Jurassic as a test, and also to massively increase terminal/internal branch lengths going into the Cretaceous (as a contrast to the main analysis). Below the plot shows a consensus tree from the 50 cal-3 iterations and you can see many notosuchians nodes are now being placed in the Jurassic period (highlighted in grey).

When looking at rates through time, notosuchians (red line, consensus from 50 trees) still show faster rates than the background (grey line, consensus 50 trees) across all crocodylomorphs for a sustained interval, including in the Cretaceous. This is despite the massive (I would suggest highly unrealistic) stretching of branch lengths and pulling origins, and fast branching, of 'basal' notosuchians nodes into the Jurassic.

Overall, we hope these additional tests show that the methods and conclusions for our work are robust, particularly to quite substantial additions and perturbations to the tree and data.

Many thanks,

The authors.